# Improvement in Processability for Injection Molding of Bisphenol-A Polycarbonate by Addition of Low-Density Polyethylene

**DOI:** 10.3390/ma16020866

**Published:** 2023-01-16

**Authors:** Yuki Kuroda, Ken-Ichi Suzuki, Genzo Kikuchi, Nantina Moonprasith, Takumitsu Kida, Masayuki Yamaguchi

**Affiliations:** 1Polymer Materials Research Laboratory, Tosoh Corporation, 1-8 Kasumi, Yokkaichi 510-8540, Mie, Japan; 2School of Materials Science, Japan Advanced Institute of Science and Technology, 1-1 Asahidai, Nomi 923-1292, Ishikawa, Japan; 3Sirindhorn International Institute of Technology, Thammasat University, 99 Moo 18, Paholyothin, Khlong Luang 12120, Thailand

**Keywords:** polymer blend, shear viscosity, injection molding, polycarbonate, low-density polyethylene

## Abstract

The rheological properties and processability at injection molding were studied for bisphenol-A polycarbonate (PC) that was modified by low-density polyethylene (LDPE) having a low shear viscosity. The LDPE addition significantly decreased the steady-state shear viscosity, especially in the high shear rate region. The decrease did not originate from slippage on the die wall but due to interfacial slippage between the PC and dispersed LDPE droplets that deformed to the flow direction to a great extent. As a result of the viscosity decrease, injection pressure largely decreased from 150 to 110 MPa with the addition only 5 wt.% of LDPE. The enhanced flowability also reduced the warpage of the molded product significantly, demonstrating that the processability at injection molding was improved by the addition of LDPE.

## 1. Introduction

There has been a great demand to reduce the weight of automobile parts, particularly as electric vehicles become more widespread. Bisphenol-A polycarbonate (PC) is one of the most important plastics for automobile parts because of its mechanical toughness [1,2,3]. Roughly speaking, the impact strength of PC is five times as high as that of acrylonitrile-butadiene-styrene terpolymer (ABS) and 50 times as high as that of polymethylmethacrylate (PMMA). Therefore, PC is used not only for transparent parts but also colored ones. It is also known that PC shows a high glass transition temperature *T_g_*, which is around 150 °C [1,2,3]. Therefore, the heat deflection temperature under 1.81 MPa is around 120–130 °C, which is much higher than those of ABS (75–90 °C) and PMMA (85–105 °C) [1,2,3]. However, the high *T_g_* of PC often results in poor flowability during injection molding due to its rapid solidification in the mold, which becomes a severe problem when producing thin parts. To overcome the problem, injection compression molding is often employed to produce thin parts, although to the detriment of cost performance [4,5]. Therefore, it is strongly required for PC to decrease shear viscosity without decreasing molecular weight that affects the mechanical toughness, such as impact strength. The addition of a lubricant has been a well-known method to improve the flow length during injection molding [6,7]. A lubricant is localized between a polymer and a metal wall, which then leads to slippage on the surface of a mold. However, it may give the product a rough surface, resulting in difficulties with the hard-coating process that are inevitable for automobile applications of PC. Adding a plasticizer that is miscible with a polymer is another general technique to decrease shear viscosity. However, it leads to a decrease in *T_g_*, i.e., poor heat resistance.

It was reported recently that the addition of polystyrene (PS) with low molecular weight reduced the shear viscosity of PC [8,9]. PS is immiscible with PC and thus the blends show phase-separated structure. Once shear flow is applied to the blend, the PS droplets, having a low shear viscosity, are deformed to the flow direction greatly. Due to the high interfacial tension between the PC and PS [10], the interfacial thickness decreases, and interfacial slippage takes place during flow [11,12,13,14,15]. This is the origin of viscosity drop [9,16]. When the interfacial tension is not so high, this leads to a thick interfacial thickness, and shear viscosity of the blend does not decrease greatly. Such a phenomenon was confirmed for the blends of polypropylene (PP) as a continuous phase and ethylene-α-olefin copolymer as dispersion [16].

This phenomenon could be applicable to injection molding, although it has not been reported yet. Since the viscosity drop will reduce the molecular orientation of a product, the warpage and anisotropy during shrinkage would be minimized. However, due to the poor thermal stability of PS, which sometimes generates volatile compounds [17] with poor mechanical toughness [18], PC/PS blends have not been employed widely.

In this study, we picked up low-density polyethylene (LDPE) as a modifier of rheological properties, because LDPE is a thermally stable material. Even when thermal degradation occurs, LDPE shows a crosslinking reaction, not chain scission [19], which is different from PP. Therefore, the blends will not generate volatile compounds as long as they are processed after a drying procedure to remove moisture. Moreover, LDPE has a high interfacial tension with PC, and the blends show phase-separated structure [18,20,21,22,23,24,25,26]. The structure and properties for the blends of PC and polyethylene have been studied by several researchers, including their rheological properties [21] and processability at injection molding [24]. To the best of our knowledge, however, no research has yet focused on interfacial slippage and its impact on the processability at injection molding. Since the addition of LDPE did not appreciably worsen the mechanical properties, such as modulus and impact strength [18,20], the modification of PC by LDPE with its low shear viscosity is interesting to consider, given its good cost performance.

In this study, besides the rheological measurements with morphological observations, we carried out injection molding to investigate the processability of PC modified with LDPE. Considering the growing concern regarding plastic recycling [27,28,29,30], for which PC usually shows poor flowability during injection molding, the results of this study should be noted.

## 2. Materials and Methods

### 2.1. Materials

As a matrix polymer, PC (Iupilon S2000; Mitsubishi Engineering Plastics, Tokyo, Japan) was used. The number- and weight-average molecular weights, evaluated via size exclusion chromatography (SEC; HLC-8020; Tosoh, Tokyo, Japan), using chloroform as a solvent with a polystyrene standard, were *M_n_* = 2.8 × 10^3^ and *M_w_* = 5.7 × 10^4^, respectively. Furthermore, a commercially available LDPE, produced by Tosoh corporation, was employed as a modifier. The number- and weight-average molecular weights, evaluated by another SEC (HLC-8321GPC/HT; Tosoh) using 1,2,4-trichlorobenzene as a solvent at 140 °C with a linear polyethylene standard, were *M_n_* = 1.2 × 10^4^ and *M_w_* = 6.2 × 10^4^, respectively. As antioxidants, tris(2,4-di-tert-butylphenyl)phosphate (Irgafos168; Ciba Specialty Chemicals, Basel, Switzerland) and pentaerythritol tetrakis(3-[3,5-di-tert-butyl-4-hydroxyphenyl]propionate) (Irganox1010; Ciba Specialty Chemicals) were added. The content was 3000 ppm per polymer materials for each antioxidant.

### 2.2. Sample Preparation

Blend samples were prepared by melt-mixing using a co-rotating twin-screw extruder (TEX30ss; Japan Steel Works, Tokyo, Japan) at 260 °C. Before mixing, the PC samples were dried at 120 °C for 3 h in a vacuum oven. The screw rotation speed was 150 rpm. The LDPE contents prepared were 0, 3, 5, 10, and 25 wt.%. The obtained blends were compression-molded into flat films under 10 MPa at 250 °C for 3 min. The films that were cooled at 30 °C by another compression machine were used for various measurements.

Moreover, pure PC and PC/LDPE (95/5) samples were dried again overnight at 100 °C under vacuum conditions, and they were subsequently injection-molded using an injection molding machine (MD100X; Ube Machinery, Ube, Japan). The processing was performed at the same conditions except for the injection pressure. The barrel/nozzle temperature and mold temperature were 280 °C and 80 °C, respectively. The injection speed was 40 mm/s. The hold pressure was 20 MPa.

### 2.3. Measurements

Morphological observations were performed using a scanning electron microscope (SEM; S-4100; Hitachi, Tokyo, Japan). The cryogenically fractured surface of compression-molded films was sputtered with Pt/Pd. For the injection-molded product, thin films were cut out with a microtome along the flow direction. Then the surface was exposed to the vapor of RuO_4_, followed by coating with osmium. The sample near the surface and that in the center, 2 mm from the surface, of the bar were used for the observations.

The temperature dependence of the dynamic tensile moduli in the solid state was investigated at 10 Hz using a dynamic mechanical analyzer (E-4000; UBM, Muko, Japan). The heating rate was 2 °C/min.

The frequency dependence of the oscillatory shear moduli in the molten state was investigated at various temperatures using a cone-and-plate rheometer (AR2000ex; TA Instruments, New Castle, DE, USA).

Steady-state shear viscosity was measured using a pressure-driven capillary rheometer (140SAS; Yasuda Seiki Seisakusyo, Nishinomiya, Japan). The temperature of the barrel and die was controlled at 250 °C. Two circular dies having the same diameter-to-length (*L*/*D*) ratio were employed, such as *L*/*D* values of 10/1 and 20/2 (mm/mm). The entrance angle of both dies was 2π.

The flexural modulus (ISO178) and notched Izod impact strength (ISO180) were measured using injection-molded samples at 23 °C.

## 3. Results and Discussion

Figure 1 shows the SEM images of the fractured surface of the blend films prepared by compression molding. It is well known that polyethylene is immiscible with PC [18,20,21,22,23,24,25,26]. All of the blends in this study also showed phase-separated structure, in which spherical LDPE droplets were dispersed in a continuous PC. As the LDPE content increased, the droplet size increased.

The angular frequency dependences of the oscillatory shear moduli, such as the storage modulus *G′* and loss modulus *G″*, are shown in Figure 2. The reference temperature was 250 °C.

Both polymers showed the rheological terminal region in the low-frequency region. Therefore, the zero-shear viscosity $\eta_{0}$, steady-state compliance $J_{e}^{0}$, and weight-average relaxation time $\tau_{W}$ were determined using the following relations [31], which are summarized in Table 1.
(1)$$\eta_{0}=\underset{\omega \to 0}{\lim}\frac{G^{\prime \prime }}{\omega }$$
(2)$$J_{e}^{0}=\underset{\omega \to 0}{\lim}\frac{G^{\prime }}{\omega^{2}}$$
(3)$$\tau_{W}=\eta_{0}J_{e}^{0}$$

It is well known that the critical capillary number of an immiscible polymer blend with sea-island structure is significantly large under shear flow, when the viscosity of dispersions is much lower than that of the continuous phase. For such a system, dispersed droplets tend to be deformed affinely without break-up during flow, as explained in previous studies [32,33,34]. Since the zero-shear viscosity of the PC was around 100 times higher than that of the LDPE in the present system, the LDPE dispersions were deformed affinely under shear flow and turned into a fibrous shape in a continuous PC phase, as demonstrated later.

The difference in steady-state compliance must have originated from the relaxation time distribution. In a classical theory, $G_{N}^{0}J_{e}^{0}$ ($G_{N}^{0}$ is the rubbery plateau modulus) of a monodispersed polymer is known to be constant, irrespective of polymer species [31,35]. Although polyethylene has a similar $G_{N}^{0}$ with PC [36,37], the LDPE had a broad molecular weight distribution with long-chain branches. As a result, $J_{e}^{0}$ of the LDPE was much larger than that of the PC.

Figure 3 shows the angular frequency dependences of the oscillatory shear moduli of the blend samples at 250 °C. The *G″* values of PC/LDPE (97/3) and (90/10) were almost identical to those of pure PC. In contrast, the *G′* values of the blends in the low-frequency region were higher than those of pure PC. This was pronounced as the LDPE content increased. It is well known that interfacial tension acting on deformed droplets in an immiscible polymer blend is responsible for the long characteristic time, and this was well summarized by the rheological emulsion model [38,39].

For the PC/LDPE (75/25) blend, both *G′* and *G″* in the high-frequency region were lower than those of pure PC. This result indicated that a part of LDPE existed continuously. Since the LDPE showed much lower viscosity than the PC, it was a reasonable result [33,40]. However, the PC must be the main polymer in the continuous phase, because the *G″* values were similar to those of pure PC.

The temperature dependence of dynamic tensile moduli, such as the storage modulus *E′* and loss modulus *E″*, is shown in Figure 4. For pure PC and LDPE films, the glass transition temperatures *T_g_*s, i.e., peak temperatures in the *E″* curve, were detected at around 155 °C for PC and −25 °C for LDPE. Correspondingly, the *E′* of pure PC dropped off sharply at *T_g_*. The LDPE showed a gradual decrease in *E′* beyond *T_g_* until the melting point due to its crystallinity. A broad *E″* peak in the temperature range from −150 to −50 °C was detected for PC, which was ascribed to the local motion [41].

For the blends, the *E″* peak ascribed to the *T*_g_ of PC was detected at the same temperature. This result demonstrated that even a small amount of LDPE was not dissolved into PC, as reported previously [18,20,21,22,23,24,25,26]. Upon the addition of 25 wt.% of LDPE, the *E″* exhibited an additional weak peak at around 100 °C. This is probably attributed to the LDPE melting, indicating that a part of the LDPE existed as a continuous phase in PC/LDPE (75/25); however, the main polymer in the continuous phase must have been PC, because the *E′* decreased greatly at *T_g_* of the PC. This result corresponded with the oscillatory shear modulus in the molten state.

The steady-state shear viscosity was measured using a pressure-driven capillary rheometer at 250 °C. As shown in Figure 5, the LDPE addition significantly reduced shear viscosity even with 3 wt.%. For the blends with 3 and 10 wt.% of LDPE, non-Newtonian behavior became pronounced compared with pure PC, demonstrating that the decreased viscosity was pronounced in the high shear rate region, which was obvious from Figure 6. This is very important for injection molding. The decrease in viscosity at high shear rates is, in contrast, not so obvious in a plasticized, i.e., miscible, system. It is well known that a viscosity decrease is pronounced in the low shear rate region for a miscible system with a low-molecular-weight compound such as plasticizer [8,9,42].

Moreover, Figure 6 also indicated that the decrease in viscosity at a high shear rate, i.e., 1000 s^−1^, was detected with a small amount of LDPE.

In this experiment, two circular dies were employed to evaluate the wall slippage. It is known that the slippage can be evaluated by the comparison of apparent shear viscosities obtained using various dies with different die lengths *L* having the same *L*/*D* ratio [43]. As shown in Figure 5, the flow curves of the blends for *L*/*D* = 10/1 and 20/2 were identical. This result demonstrated that the viscosity drop did not originate from slippage on the die wall. As reported previously [9], the interfacial slippage between PC and LDPE must occur, because the interfacial thickness significantly decreased due to high interfacial tension, i.e., poor adhesive strength [12,13,14,15,43,44]. Moreover, the huge difference in shear viscosity provided an enlarged interfacial area between PC and LDPE as a result of large deformation of the LDPE droplets [32,33], which accelerate interfacial slippage. This must be the origin of the decrease in viscosity observed in the present system.

We also performed injection molding to investigate the effect of the LDPE addition on the processability using pure PC and PC/LDPE (95/5) under the same processing conditions. At first, the structure in an injection-molded product of PC/LDPE (95/5) was observed via SEM. Samples were cut out from the bar with 4 mm thickness; one cut was near the surface (skin region) and the other was in the center (core region).

As shown in Figure 7, phase-separated structure was clearly detected, in which the white area denoted LDPE droplets. In the skin region, i.e., high shear rate region, LDPE droplets were deformed greatly to the flow direction. As a result, a large interfacial area was provided, which pronounced the interfacial slippage between PC and LDPE, as revealed in previous studies [8,9,16]. In contrast, spherical LDPE droplets were observed in the core. The droplets seemed to be aligned in a row to the flow direction, suggesting that prolonged droplets disintegrated into smaller ones through Rayleigh disturbance during cooling [33]. This is reasonable, since it took a longer time for solidification to occur in the core of a bar with 4 mm thickness. Before filling in the mold, therefore, the dispersions must have had a large interfacial area, leading to a decrease in viscosity due to slippage.

The injection-molded products of PC and PC/LDPE (95/5) are shown in Figure 8. Although PC/LDPE (95/5) was opaque due to light scattering, which originated from a large difference in the refractive indices, no bubbles were found in the product. This result supported the finding that volatile compounds were not generated at the processing temperature of PC. Moreover, it should be noted that the warpage was significantly reduced by the addition of LDPE. The degree of warpage, defined in Figure 8b, is summarized in Table 2 with the data of injection pressure. It was found that the injection pressure was significantly reduced from 150 to 110 MPa with an addition of 5wt.% of LDPE; this demonstrated that flowability certainly improved at injection molding. The improved flowability was responsible for the reduced warpage observed [4,5,45,46,47] and made it possible to obtain a thin and/or large product with reduced anisotropy in shrinkage. Considering that the viscosity drop became pronounced under high shear stress conditions, low-temperature-processing should be available for PC.

The mechanical properties such as flexural modulus and notched Izod impact strength were as follows: 2240 MPa and 82 kJ/m^2^ for PC and 2040 MPa and 76 kJ/m^2^ for PC/LDPE (95/5), respectively. The results demonstrated that these mechanical properties were not affected greatly. Although the impact strength was not enhanced to the same degree as ABS and core-shell latex elastomer [48,49,50], the impact on the processability should be noted.

## 4. Conclusions

The impact of the LDPE addition on the rheological properties and processability for injection molding of PC was studied. When the LDPE content was lower than or equal to 10 wt.%, LDPE existed as a dispersed rather than a continuous phase. The steady-state shear viscosities decreased greatly, even with 3 wt.% of LDPE; this was pronounced in the high shear rate region. The viscosity decrease did not originate from wall slippage. The enlarged interfacial area provided by the large viscosity difference, with poor adhesive strength between PC and LDPE, led to the interfacial slippage that was responsible for the decrease in viscosity. Moreover, it should be noted that the addition of a small amount of LDPE remarkably reduced the warpage of an injection-molded product and decreased the injection pressure without generating any volatile compounds. This technique will be available to obtain thin and/or large PC products with injection molding, which are valuable for industry.

## Figures and Tables

**Figure 1 materials-16-00866-f001:**
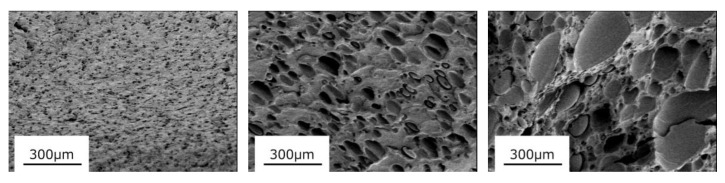
SEM images of the fractured surfaces of compression-molded films; (**left**) PC/LDPE (97/3), (**center**) PC/LDPE (90/10), and (**right**) PC/LDPE (75/25).

**Figure 2 materials-16-00866-f002:**
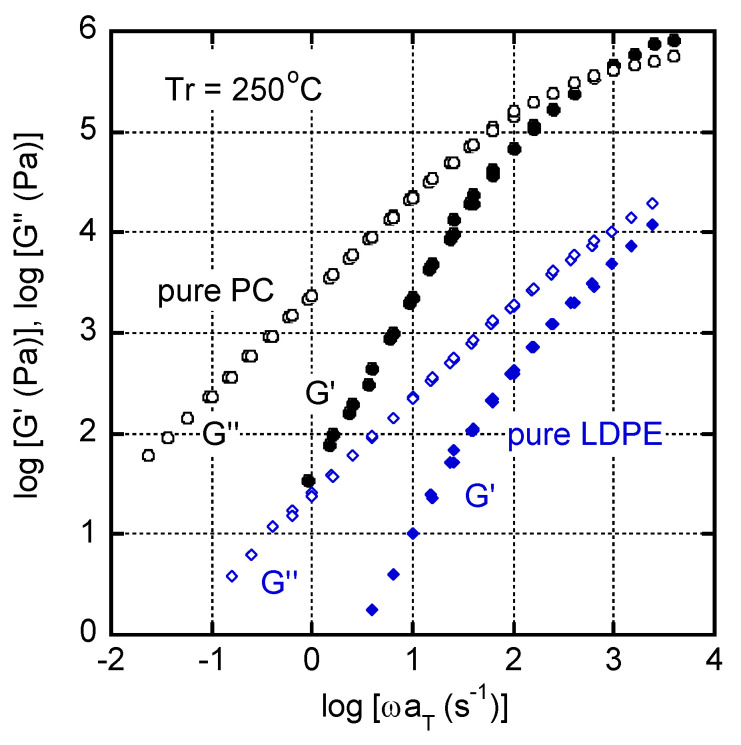
Master curves of oscillatory shear moduli such as the storage modulus *G′* (closed symbols) and loss modulus *G″* (open symbols) at 250 °C for PC (circles) and LDPE (diamonds).

**Figure 3 materials-16-00866-f003:**
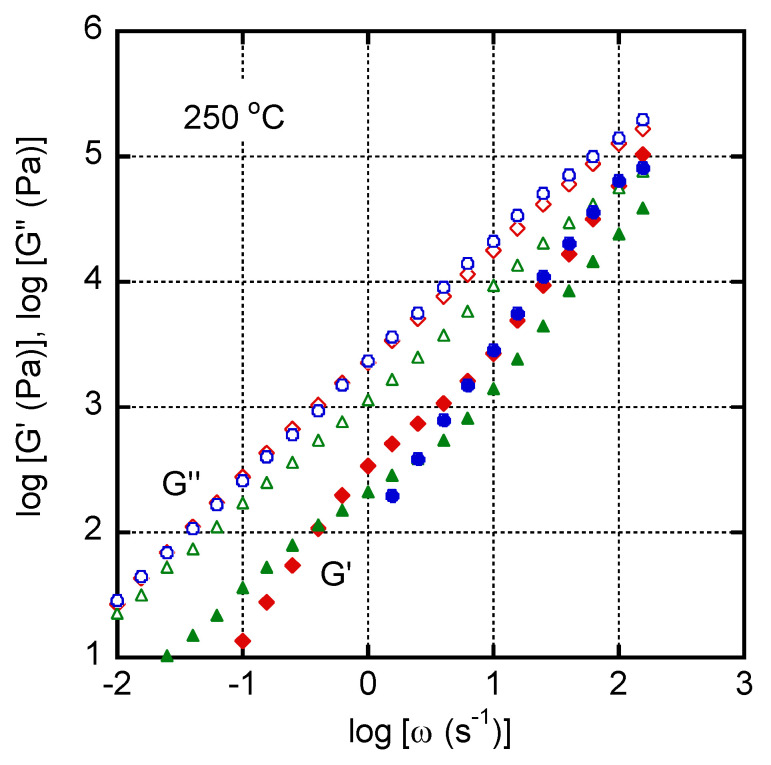
Angular frequency dependences of oscillatory shear moduli such as storage modulus *G′* (closed symbols) and loss modulus *G″* (open symbols) at 250 °C for PC/LDPE (97/3) (circles), PC/LDPE (90/10) (diamonds), and PC/LDPE (75/25) (triangles).

**Figure 4 materials-16-00866-f004:**
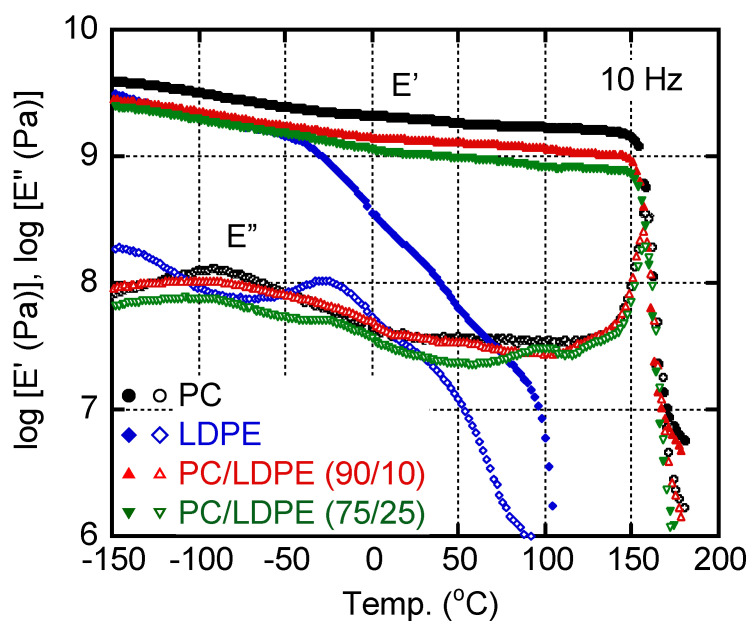
Temperature dependence of tensile storage modulus *E′* (closed symbols) and loss modulus *E″* (open symbols) at 10 Hz for PC (circles), LDPE (diamonds), PC/LDPE (90/10) (triangles), and PC/LDPE (75/25) (inverted triangles).

**Figure 5 materials-16-00866-f005:**
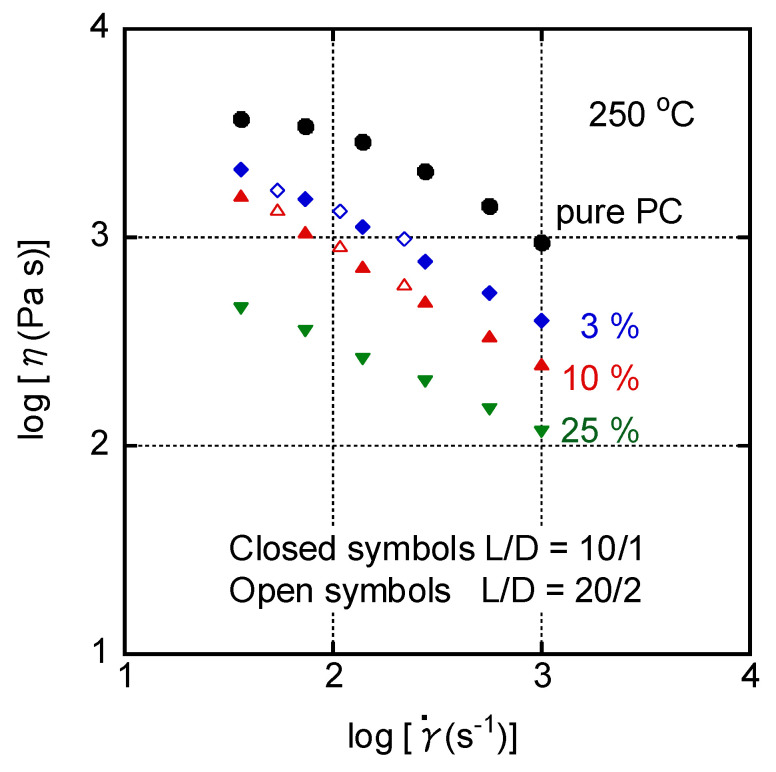
Steady-state shear viscosity *η* plotted against the shear rate $\dot{\gamma }$ at 250 °C for PC (circles), PC/LDPE (97/3) (diamonds), PC/LDPE (90/10) (triangles), and PC/LDPE (75/25) (inverted triangles) obtained by two capillary dies having the same L/D ratio. Closed symbols represent *L/D =* 10/1 and open symbols represent *L/D* = 20/2.

**Figure 6 materials-16-00866-f006:**
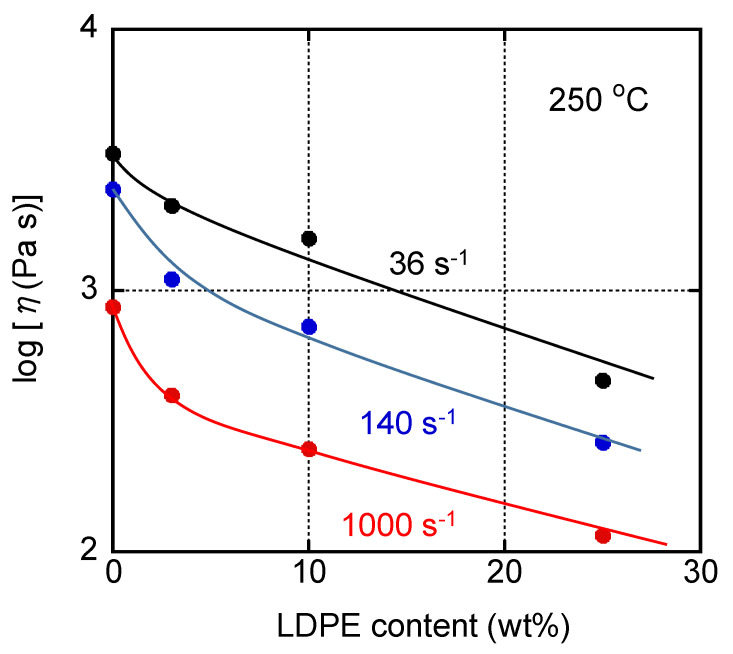
Steady-state shear viscosity *η* plotted against the LDPE content at various shear rates at 250 °C. The measurements were carried out using a circular die with *L/D* = 10/1.

**Figure 7 materials-16-00866-f007:**
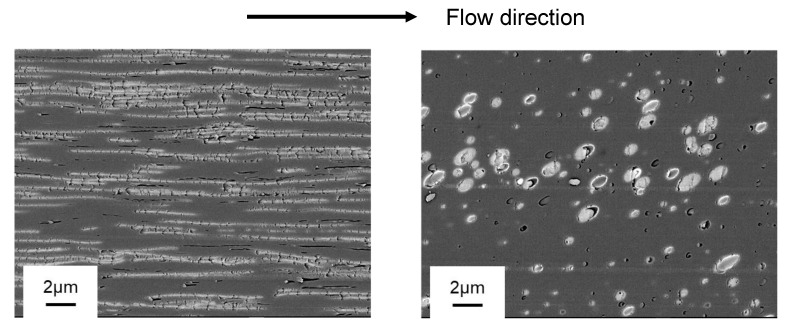
SEM images in an injection-molded product of PC/LDPE (95/5); the (**left**) shows the skin region, and the (**right**) shows the core region. The arrow indicates the flow direction.

**Figure 8 materials-16-00866-f008:**
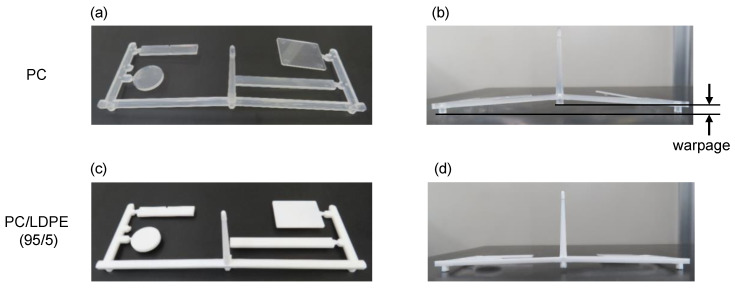
Injection-molded products of (**a**,**b**) PC and (**c**,**d**) PC/LDPE (95/5).

**Table 1 materials-16-00866-t001:** Rheological parameters of pure polymers.

Materials	Zero-Shear Viscosity (Pa s)	Steady-State Compliance (Pa^−1^)	Weight-Average Relaxation Time (s)
PC	2.3 × 10^3^	7.5 × 10^−5^	1.7 × 10^−2^
LDPE	2.4 × 10^1^	2.0 × 10^−4^	4.8 × 10^−3^

**Table 2 materials-16-00866-t002:** Injection pressures and warpage values.

Materials	Pressure (MPa)	Warpage (mm)
PC	1.5 × 10^2^	8.3
PC/LDPE (95/5)	1.1 × 10^2^	2.6

## Data Availability

Not applicable.

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
