# Peer review of "Improvement in Processability for Injection Molding of Bisphenol-A Polycarbonate by Addition of Low-Density Polyethylene"

_materials, 2023, doi:10.3390/ma16020866_

Round 1

Reviewer 1 Report

1. Introduction section

-Try to avoid expresssions such as "excellent mechanical properties", "good heat resistance", "poor heat resistance". This adjectives are subjective, and in the case of mechanical properties ¿wich ones? It is bette to use semiquantitative adjectives as "high-low" or similar ones. Revise these expresssions along the text.

-Last paragraph: "we picked up LDPE as a modifier or rheological properties, because is a thermally stable material". Is it the only reason? Why not choosing other kind of polymeric stable materials. More explanation is needed to justify the selection of LDPE.

2.2 Sample preparation

"PC and PC blends were dried"...where? dry oven? dehumidifier? Please, more detail

3. Results and discussion

Page 4: "As increasing LDPE content, the droplet size increased". It should be helpful to identify in some way in FIgure 1 what is taken as a LDPE droplet. Also to whow the turning into fibrous shape as you mention in page 6.

References

Nr. 6. Missed initial name of Hatzikiriakos.

Nr 8. 170 is in bold.

Nr 11. Year 1996 is not written in bold

Nr 35. Erase (2016) at the end.

Revise also that all the references end with a dot, in some cases is missed.

Author Response

We appreciate the reviewing process and thoughtful comments on our manuscript (Manuscript ID materials-2123928). All of them are important to polish our manuscript. The response to each inquiry is as follows;

  1. Introduction section

-Try to avoid expresssions such as "excellent mechanical properties", "good heat resistance", "poor heat resistance". This adjectives are subjective, and in the case of mechanical properties ¿which ones? It is better to use semiquantitative adjectives as "high-low" or similar ones. Revise these expresssions along the text.

[our response]

Thank you for reviewing process. We understand the suggestion. The expressions were changed in the revised manuscript as follows.

[revised]

Page 3, 5th line

Roughly speaking, the impact strength of PC is 5 times as high as that of acrylonitrile-butadiene-styrene terpolymer (ABS), and 50 times as high as that of polymethylmethacrylate (PMMA). Therefore, PC is used not only for transparent parts but colored ones. It is also known that PC shows a high glass transition temperature Tg, which is around 150 °C [1-3]. Therefore, the heat deflection temperature under 1.81 MPa is around 120-130 °C, which is much higher than those of ABS (75-90 °C) and PMMA (85-105 °C) [1-3].

Page 16, 2nd line

The mechanical properties such as flexural modulus and notched Izod impact strength were as follows:

-Last paragraph: "we picked up LDPE as a modifier or rheological properties, because is a thermally stable material". Is it the only reason? Why not choosing other kind of polymeric stable materials. More explanation is needed to justify the selection of LDPE.

[our response]

We added other reasons to select LDPE. Thank you for the comment.

[revised]

Page 4, 9th line from the bottom

Even when thermal degradation occurs, LDPE shows a crosslinking reaction, not chain scission [19], which is different from PP. Therefore, the blends will not generate volatile compounds as long as they are processed after a drying procedure to remove moisture. Moreover, LDPE has a high interfacial tension with PC, and the blends show phase-separated structure [18,20-26].

Page 5, 1st line

Since the addition of LDPE did not appreciably worsen the mechanical properties, such as modulus and impact strength [18,20], the modification of PC by LDPE with its low shear viscosity is interesting to consider, given its good cost performance.

2.2 Sample preparation

"PC and PC blends were dried"...where? dry oven? dehumidifier? Please, more detail

[our response]

We added in the revised manuscript.

[revised]

Page 6, 4th line

Before mixing, the PC samples were dried at 120 °C for 3 h in a vacuum oven.

Page 6, 9th line

Moreover, pure PC and PC/LDPE (95/5) samples were dried again overnight at 100 °C under vacuum conditions,

  1. Results and discussion

Page 4: "As increasing LDPE content, the droplet size increased". It should be helpful to identify in some way in Figure 1 what is taken as a LDPE droplet. Also to how the turning into fibrous shape as you mention in page 6.

[our response]

The minor phase is regarded as LDPE, because the volume fraction of LDPE is lower than that of PC. In the case of PC/LDPE (70/30), the decision is not so easy because a low-viscous component may be a continuous phase with a small amount. However, the dynamic mechanical properties in both solid and molten states suggested that the main polymer in the continuous phase must be PC. We added the following sentences to clarify it. Regarding the mechanism to deform into fibrous shape, we added more explanation with references [32-34] of previous studies.

[revised]

Page 9, 13th line from the bottom

For such a system, dispersed droplets tend to be deformed affinely without break-up during flow, as explained in previous studies [32-34]. Since the zero-shear viscosity of the PC was around 100 times higher than that of the LDPE in the present system, the LDPE dispersions were deformed affinely under shear flow and turned into a fibrous shape in a continuous PC phase, as demonstrated later.

Page 10, 2nd line from the bottom

However, the PC must be the main polymer in the continuous phase, because the G” values were similar to those of pure PC.

Page 12, 1st line

however, the main polymer in the continuous phase must have been PC, because the E’ decreased greatly at Tg of the PC.

References

Nr. 6. Missed initial name of Hatzikiriakos.

Nr 8. 170 is in bold.

Nr 11. Year 1996 is not written in bold

Nr 35. Erase (2016) at the end.

Revise also that all the references end with a dot, in some cases is missed.

[our response]

Thank you so much for the careful check and kind suggestions. We corrected them all.

Reviewer 2 Report

In this work, the authors have investigated the rheological and processability properties of PC modified by LDPE. 

The authors found out that LDPE reduced the steady-state shear viscosity owing to the interfacial slippage between PC and LDPE, followed by decreased injection pressure. They also claim that warpage were reduced by the advantage of the enhanced flowability. 

- First of all, the authours should need extensive language improvement to meet journal article's standard. 

- The experimental results seems to be too obvious to predict, and do not convey any novelty nor breakthrough in the research. Many researchers have already done such of the many works since early 1950s regarding polymer blending and injection molding. 

- In Fig. 7, the PC/LDPE showed less warpage than PC. The warpage results from residual stresses on the sample. The authors should investigate their residual stresses in order to explore the relationship between processabiity and warpage.

- Although the authors have investigated their rheological properties, I do not find any good reasons for this work to be published in MATERIALS journal.

Author Response

We appreciate the reviewing process on our manuscript (Manuscript ID materials-2123928). We checked them carefully. Here are our responses.

In this work, the authors have investigated the rheological and processability properties of PC modified by LDPE. 

The authors found out that LDPE reduced the steady-state shear viscosity owing to the interfacial slippage between PC and LDPE, followed by decreased injection pressure. They also claim that warpage were reduced by the advantage of the enhanced flowability. 

[our response]

Thank you for the reviewing process.

- First of all, the authours should need extensive language improvement to meet journal article's standard. 

[response]

The revised manuscript was checked and corrected by the MDPI English editing service.

- The experimental results seems to be too obvious to predict, and do not convey any novelty nor breakthrough in the research. Many researchers have already done such of the many works since early 1950s regarding polymer blending and injection molding. 

 [our response]

Although many papers have been published on the theological properties for polymer blends, our system is different from them by the following three points (especially third one has a high novelty); (1) The shear viscosity of the minor fraction is significantly lower than that of the matrix. Such blends were not studied extensively, because it is not preferred to obtain fine morphology. Once the minor fraction is miscible with the matrix, the system is different from ours. (2) The interfacial tension is high. Usually, such a system is avoided to be employed because it gives coarse morphology. (3) The slippage inside of materials, i.e., between phases, is quite unique, although blend systems showing slippage on the wall surface were reported previously. For example, a lubricant induces the wall slippage, not slippage between phases. In fact, rheological properties of PC/LDPE have not been studied so much (previous researches were cited in the revised version), Only exception must be the interfacial slippage in the coextruded films, which were firstly studied by Macosko et al. However, their studies were focused on the multilayered film, not phase-separated blends. Slippage occurred between layers, not in the blends.

These points were insisted in the revised manuscript to appeal what is new in this paper.

[revised]

Page 3, 8th line from the bottom

The addition of a lubricant has been a well-known method to improve the flow length during injection molding [6,7]. A lubricant is localized between a polymer and a metal wall, which then leads to slippage on the surface of a mold. However, it may give the product a rough surface, resulting in difficulties with the hard-coating process which are inevitable for automobile applications of PC. Adding a plasticizer, that is miscible with a polymer, is another general technique to decrease shear viscosity. However, it leads to a decrease in Tg, i.e., poor heat resistance.

Page 4, 6th line

When the interfacial tension is not so high, this leads to a thick interfacial thickness, and shear viscosity of the blend does not decrease greatly. Such a phenomenon was confirmed for the blends of polypropylene (PP) as a continuous phase, and ethylene-a-olefin copolymer as dispersion [16].

Page 4, 6th line from the bottom

Moreover, LDPE has a high interfacial tension with PC, and the blends show phase-separated structure [18,20-26]. The structure and properties for the blends of PC and polyethylene have been studied by several researchers, including their rheological properties [21] and processability at injection molding [24]. To the best of our knowledge, however, no research has yet focused on interfacial slippage and its impact on the processability at injection molding.

Page 5, 1st line

Since the addition of LDPE did not appreciably worsen the mechanical properties, such as modulus and impact strength [18,20], the modification of PC by LDPE with its low shear viscosity is interesting to consider, given its good cost performance.

Page 12, 6th line

For the blends with 3 and 10 wt.% of LDPE, non-Newtonian behavior became pronounced compared with pure PC, demonstrating that the decreased viscosity was pronounced in the high shear rate region, which was obvious from Figure 6. This is very important for injection molding. The decrease in viscosity at high shear rates is, in contrast, not so obvious in a plasticized system. It is well known that a viscosity decrease is pronounced in the low shear rate region for a miscible system with a low-molecular-weight compound such as plasticizer [8,9,42].

- In Fig. 7, the PC/LDPE showed less warpage than PC. The warpage results from residual stresses on the sample. The authors should investigate their residual stresses in order to explore the relationship between processabiity and warpage.

[our response]

Thank you for the good comment. In fact, birefringence is a conventional way to check the residual stress, although we can’t measure it due to the light scattering of the blend sample. We found that the injection pressure was greatly reduced under the same processing condition, suggesting that the residual stress decreased. We believe that it is easily predictable for the readers in this field.

- Although the authors have investigated their rheological properties, I do not find any good reasons for this work to be published in MATERIALS journal.

[response]

Again, we insisted on the novelty as shown in the reply to the first comment. Hope the reviewer understand the significance of this study. The information obtained by this study is going to be employed in industry because it is quite effective to increase the flow length at injection-molding.

Reviewer 3 Report

The study presented is interesting, but lacks in depth and has an at least questionable interpretation. In general, the following points need to be addressed in revision:

- Literature on PC/PE blends is missing completely in the introduction - I admit that there's not too much, but papers like  https://doi.org/10.1002/mame.200400145  and https://doi.org/10.1002/pen.760301706 (the latter explicitely dealling with rheology) need to be discussed. 

- While it may be correct, that PC/PS blends have not found wide application, the well established PC/ABS blends should be mentioned (see e.g. the review https://doi.org/10.1002/adv.1994.060130401 )

- The rheology of the blends must be compared to the result of a simple mixing rule in order to verify or falsify the suggested interfacial slippage effect. 

- It would be good to present mechanical and optical properties of the resulting blends in order to see to which extent they still can be used in the original PC application. 

Some detail corrections also are necessary:

- page 2 "after a drying process" instead of "after dry process"

- page 6 "almost identical to those of" instead of "almost the same with those of"

- page 8: The sentence "This is very important for injection molding but not so obvious in a plasticized system." is unclear and misleading - plasticizing a polymer is defined as adding a low molecular weight substance, which is miscible. Adding a second immiscible polymer is blending. 

Author Response

We appreciate the reviewing process and thoughtful comments on our manuscript (Manuscript ID materials-2123928). All of them are important to polish our manuscript. The response to each inquiry is as follows;

The study presented is interesting, but lacks in depth and has an at least questionable interpretation. In general, the following points need to be addressed in revision:

[our response]

Thank you very much for reviewing process and kind suggestions.

- Literature on PC/PE blends is missing completely in the introduction - I admit that there's not too much, but papers like  https://doi.org/10.1002/mame.200400145  and https://doi.org/10.1002/pen.760301706 (the latter explicitely dealling with rheology) need to be discussed. 

[our response]

We added several papers on PC/LDPE including those suggested by the reviewer. Some discussion based on the previous researches was made as follows:

[revised]

Page 4, 6th line from the bottom

Moreover, LDPE has a high interfacial tension with PC, and the blends show phase-separated structure [18,20-26]. The structure and properties for the blends of PC and polyethylene have been studied by several researchers, including their rheological properties [21] and processability at injection molding [24]. To the best of our knowledge, however, no research has yet focused on interfacial slippage and its impact on the processability at injection molding. Since the addition of LDPE did not appreciably worsen the mechanical properties, such as modulus and impact strength [18,20], the modification of PC by LDPE with its low shear viscosity is interesting to consider, given its good cost performance.

- While it may be correct, that PC/PS blends have not found wide application, the well established PC/ABS blends should be mentioned (see e.g. the review https://doi.org/10.1002/adv.1994.060130401 )

 [our comment]

Thank you. We added some comments on ABS blends in the revised manuscript, which are important in industry as suggested by the reviewer. Of course, the review paper suggested by the reviewer was included.

[revised]

Page 16, 5th line

Although the impact strength was not enhanced to the same degree as ABS and core-shell latex elastomer [48-50], the impact on the processability should be noted.

- The rheology of the blends must be compared to the result of a simple mixing rule in order to verify or falsify the suggested interfacial slippage effect. 

[our response]

We believe that the simple mixing rule does not contain any scientific meaning especially in the non-Newtonian region. However, as the reviewer commented, we added a new figure (Figure 6), i.e., shear viscosity as a function of blend ratio at various shear rates, which will insist the viscosity drop more clearly.

[revised]

Adding Figure 6

Page 12, 6th line

For the blends with 3 and 10 wt.% of LDPE, non-Newtonian behavior became pronounced compared with pure PC, demonstrating that the decreased viscosity was pronounced in the high shear rate region, which was obvious from Figure 6. This is very important for injection molding. The decrease in viscosity at high shear rates is, in contrast, not so obvious in a plasticized system.

Page 13, 1st line

Moreover, Figure 6 also indicates that the decrease in viscosity at a high shear rate, i.e., 1000 s-1, was detected with a small amount of LDPE.

- It would be good to present mechanical and optical properties of the resulting blends in order to see to which extent they still can be used in the original PC application. 

[our response]

We added the results of the mechanical properties in the revised version including the measurement methods. For the optical properties, we just commented that the LDPE blend was opaque due to light scattering.

[revised]

Page 7, 12th line

The flexural modulus (ISO178) and notched Izod impact strength (ISO180) were measured using injection-molded samples at 23 °C.

Page 15, 11th line from the bottom

Although PC/LDPE (95/5) was opaque due to light scattering, which originated from a large difference in the refractive indices, …

Page 16, 2nd line

The mechanical properties such as flexural modulus and notched Izod impact strength were as follows: 2240 MPa and 82 kJ/m2 for PC, and 2040 MPa and 76 kJ/m2 for PC/LDPE (95/5), respectively. The results demonstrated that these mechanical properties were not affected greatly.

Some detail corrections also are necessary:

- page 2 "after a drying process" instead of "after dry process"

- page 6 "almost identical to those of" instead of "almost the same with those of"

[our response]

Thank you. We revised them. In this revised version, a professional editing person in MDPI edited our English.

- page 8: The sentence "This is very important for injection molding but not so obvious in a plasticized system." is unclear and misleading - plasticizing a polymer is defined as adding a low molecular weight substance, which is miscible. Adding a second immiscible polymer is blending. 

[our response]

Thank you for the comment. We revised it to avoid misunderstanding.

[revised]

Page 12, 9th line

This is very important for injection molding. The decrease in viscosity at high shear rates is, in contrast, not so obvious in a plasticized, i.e., miscible, system.

Round 2

Reviewer 2 Report

The language of revised manuscript is now well corrected. Although this manuscript is only focused on their rheological characteristics and processability of the materials, they found a basic knowledge on the relationship between the processability and flowability of the injection molded materials. I recommend this manuscript to be published.

Reviewer 3 Report

The revisions have made the paper fully acceptable from my perspective.